# Estimating the Impact of COVID-19 Pandemic on Customers' Dining-Out Activities in South Korea

Bowon Suh [1], Shinyoung Kang [2] and Hyeyoung Moon [3,*]

1   Department of Food Service & Culinary Management, Kyung Hee Cyber University, Seoul 02447, Korea; bowonsuh@khcu.ac.kr
2   Department of Airline Service, Kwangju Women's University, Gwangju 62396, Korea; sykang@kwu.ac.kr
3   Institute of Symbiotic Life-TECH (Technology, Ecology, Culture, Human), Yonsei University, Seoul 03722, Korea
*   Correspondence: hye-moon@hanmail.net; Tel.: +82-10-2223-3431

**Abstract:** This study classified the types of dining-out activities into three categories: visiting restaurants, using delivery services, and using take-out services to understand how customers' various dining-out activities were carried out during the COVID-19 pandemic. The study used the Theory of Planed Behavior (TPB) model to analyze the structural relationship between the main factors and three dining-out activities. An online survey method was used to distribute and collect survey link addresses through respondents' SNS and e-mail and a data analysis was performed on the final 429(85.8%) effective samples. A paired t-test and structural equation modeling (SEM) were used to investigate customers' dining-out activities. This study is of significant contribution in that it compared and analyzed customers' various dining-out activities using the TPB model, laid the theoretical foundation for related research, and suggested ways to help related industry workers establish marketing strategies under the pandemic.

**Keywords:** dining-out activity; COVID-19; theory of planned behavior; comparative method

## 1. Introduction

This study focused on understanding how customers' various dining-out activities were carried out during the COVID-19 pandemic. Humanity has experienced numerous new diseases so far. However, unlike any previous disease, COVID-19 has had a significant adverse effect on all people's daily lives and the global industrial economy [1]. Following the rapid spread of COVID-19, the World Health Organization (WHO) declared a pandemic around the world [2]. To prevent the rapid and widespread spread of COVID-19, countries around the world implemented travel restrictions, street or city closures, and applied masks and social distancing to prevent infection [1,3].

However, these anti-infection and anti-proliferation policies have consequently had a devastating impact on the global industrial economy, particularly the hospitality and tourism industries [4]. Traveling abroad has become difficult due to restrictions on movement and city closures, and travel industry performance in major cities around the world has decreased by 70 to 90% year-on-year [5]; this has caused very serious business deterioration during the COVID-19 pandemic, especially on the food service industry [4]. According to a report by the National Restaurant Association, restaurant sales across the U.S. fell 47% as of March 2020 and about 3% of restaurants closed; the food service industry suffered a loss of about $120 billion in sales in the three months following the World Health Organization's pandemic declaration [6]. In South Korea, it is said that it succeeded in preventing the spread of infections of COVID-19 without the need for street and city closures, but the food service industry has not avoided the bad effects of COVID-19. Regarding the operation of restaurants, the government implemented various preventive measures such as spacing tables, installing partitions between tables, limiting the number of customers entering the

restaurant, and wearing masks before and after meals [7]. In addition, restaurant business hours were restricted during the social distancing phase, resulting in a significant decrease in sales in the food service industry [8].

The global turmoil caused by COVID-19 has also affected customers' daily lives and customer behavior [9]. The factors that customers value when purchasing products and services have changed significantly since the outbreak of COVID-19, and these changes will have a significant impact on the entire industry [10]. Pandemics are expected to continue over the long term, which will have a profound impact on customers' consumption behavior [11], thereby affecting the industry's marketing strategy. Therefore, research on customer behavior in the hospitality industry under the COVID-19 pandemic is essential [12]. Researchers have conducted research on COVID-19 in the fields of hospitality and the dining-out industry, but customer research is still insufficient [13]. Some customer-related studies have only investigated customer awareness and perceptions such as overseas travel perception and satisfaction analysis [14], and risk perceptions about restaurant food [15], but research on comparative analysis is still very lacking in identifying various specific behaviors of customers in relation to COVID-19.

In this study, the types of dining-out activity were classified into three categories: visiting restaurants, using delivery services, and using take-out services. Firstly, the degree of use of the three dining-out activities will be compared based on data from before and after COVID-19. Next, using the Theory of Planned Behavior (TPB) model, the study will compare and analyze the structural relationship between attitude, subjective norm, perceived behavioral control, and behavioral intention for each type of dining-out activity, and finally, investigate customers' dining-out activities during the pandemic.

## 2. Literature Review and Research Question

### 2.1. Pandemic and Food Service Industry in South Korea

COVID-19, a new type of coronavirus disease that occurred in 2019, has caused unexpected confusion around the world due to difficulties in control, rapid contagion, and limitations in treatment [16]. According to the Ministry of Health and Welfare of South Korea, as of 23 June 2021, a total of 178,350,157 patients were infected with COVID-19 worldwide, with about 3,870,443 deaths, and 152,545 people were infected and 2007 died in South Korea [17].

Considering the characteristics of such fast, infectious, and high-fatality diseases, countries around the world have chosen social distancing as the main measure for COVID-19 prevention [18]. Social distancing can be defined as self-isolation, prohibition of group gatherings, restrictions on urban travel, and non-important interruption of trade [19]. COVID-19 also made people nervous, especially about food safety issues [15], because food has properties that make it difficult to evaluate safety accurately before eating [20]. In fact, some media organizations and several research institutions reported that foods cooked by people infected with COVID-19 could infect people who ate them, which increased customer anxiety. Some customers have also begun to raise concerns about the safety of food supplies [15]. This social distancing and anxiety caused by COVID-19 has had a significant negative impact, especially on the food service industry. In South Korea, about 30,000 out of 420,000 restaurants nationwide closed from January to August 2020, and more than 4000 restaurants closed temporarily. In particular, in September 2020, with the announcement of phase 2.5 of social distancing, all restaurants were banned from operating after 9 p.m., and the restaurant industry has taken heavy damage [8]. However, on the other hand, the costs of cooking food at home and delivery food services have increased dramatically, with online food consumption increasing by 43% and delivery food consumption by 79% in May 2020 [21].

*2.2. Customer's Dining-Out Activity*

2.2.1. Visiting Restaurants during the Pandemic

COVID-19, which continues to this day, has had a very bad effect on the lives of people and economies around the world [22,23]. Among them, the most critically affected industries are the food and restaurant industries [24]. One report also reported that revenues of restaurants fell by 85 percent [25]. In particular, it can be interpreted that the reason why sales of restaurants have decreased so much is that customers do not think it is safe to eat inside the restaurant. In fact, according to a survey conducted in March 2020, 89 percent of customers think food that has been packaged at grocery stores or cooked at home is safer than food eaten at restaurants [26]. To reduce this anxiety among customers and reduce the risk of contagion of COVID-19, restaurants have applied quarantine policies to their operations [27]. All staff and all customers must wear masks inside the restaurant, social distancing signs are displayed in the restaurant, and customers are allowed to sit apart from each other. Nevertheless, customers are still anxious about eating in restaurants, and only 40 percent of customers go to bars and restaurants to eat, according to a survey conducted in Australia [28]. It is not much different in South Korea. South Korea's government provided emergency disaster relief funds to all its citizens in June 2020 to revitalize the local economy. However, eating in a restaurant is likely to cause COVID-19 infection during meals because people must take off their masks and eat in a crowded place; then, customers are reluctant to eat in restaurants, and sales of restaurant have not risen. Therefore, restaurant companies will have to find ways for customers to eat with confidence in order to solve these problems [7,29].

Several researchers have studied the impact of COVID-19 on restaurant visits by customers. Banerjee et al. [30] studied how COVID-19 affects the management of restaurants from fast-food restaurants to full-service restaurants in urban and rural areas. The results found that, since COVID-19, the number of people visiting restaurants in both urban and rural areas has decreased significantly, and urban cases have more than doubled those of rural areas. In addition, the overall rate of restaurant visits in rural areas decreased, but fast-food restaurant visits increased. Kim and Lee [31] investigated how COVID-19 affects customer preferences for private dining table preparation and private dining facilities in restaurants. According to the investigation, customers who perceive COVID-19 as a high-level threat to their health appreciated the presence of private dining tables in the restaurant, and prefer a restaurant with private dining tables over a restaurant without private dining tables. Dedeoğlu and Boğan [32] determined that how the customer's intention to visit luxury restaurants is affected by factors under the COVID-19 pandemic. In addition, the study investigated how risk awareness of COVID-19 and government confidence play a role in customers' intentions to visit restaurants as well as impact factors. Results showed that social activity and quarantine effects have a positive impact on customers' intention to visit luxury restaurants, and that government confidence and risk awareness of COVID-19 play a moderating role in the relationship between intention to visit restaurants and impact factors.

2.2.2. Using Delivery Services during the Pandemic

Health and economic systems around the world are suffering greatly from COVID-19 [33]. Many countries have imposed social distancing and several levels of lockdown, leaving many restaurants closed [24]. Non-contact with other people has been found to be a major epidemic prevention method, and the restaurant industry has begun to apply food delivery services using food service websites or online food-ordering platforms as a strategy to boost sales [34]. Customers who know that non-contact with others can reduce their risk of contracting the virus also think using food delivery services is safer than visiting restaurants. For this reason, in the current COVID-19 era, delivery services grew to be a major source of revenue for restaurant companies [35]. Large chain restaurants with large capital set up online delivery ordering systems using their websites, and delivery services were started not only from restaurant companies, but also from luxury hotels and

hotel restaurants [34,36]. Furthermore, in order to adapt and overcome this situation, in response to COVID-19, companies have also created a delivery service that leaves food on the customer's doorstep to prevent direct contact with the deliverer. These delivery services are expanding not only to food delivery, but also to daily necessities due to the advantages of effectively maintaining infection prevention during the COVID-19 pandemic [37].

Several studies regarding food delivery services during the pandemic have been conducted, as follows: Yang et al. [38] analyzed customers' reviews of the online-to-offline (O2O) platform during the COVID-19 pandemic in China and investigated what customers value when using the O2O platform. Results showed that customers value the taste, freshness, and brand reliability of food when using the O2O platform. Zhao and Bacao [37] studied whether customers who have experience using food delivery apps (FDAs) during the COVID-19 pandemic intend to continue using FDAs and what factors affect the customer's willingness to use. The study found that customers intended to continue using FDAs; satisfaction with FDAs, technical suitability and reliability of apps, and social situations positively affect the continuous use of FDAs during the COVID-19 pandemic. Kim et al. [39] identified factors that affect customers' behavioral intentions to use food delivery services utilizing drones in relation to COVID-19. The results have shown that perceived innovation in technology has a positive impact on attitudes toward food delivery services using drones. Attitudes, subjective norms, and perceived behavioral control also have a positive impact on behavioral intention to use food delivery services using drones.

### 2.2.3. Using Take-Out Services during the Pandemic

Since the beginning of the COVID-19 pandemic, governments around the world have imposed restrictions on on-site consumption in pubs, bars, cafés, and restaurants as a way to prevent the spread of virus. This restriction has resulted in a devastating impact on the food service industry and caused sharp declines in sales. In particular, unlike general restaurants, coffee shop owners have been suffering serious business difficulties as the level of distancing is more strictly applied and eating and drinking is prohibited in the coffee shops. Most coffee shops continued their service through non-face-to-face methods such as take-out and drive-thru services. Furthermore, according to HOTEL&RESTAURANT [8], restaurants unable to provide breakfast buffet services provided a Grab and Go service, which is a form of take-out service, allowing guests to consume food in their guest room [8]. Likewise, during the pandemic, many customers voluntarily or forcibly chose take-out services in various ways. One of the largest delivery application companies in South Korea, Baemin, reorganized the growth the of take-out market during the pandemic and added a new 'take-out' service tab to the main screen [40]. Take-out food is defined as cooked food which you buy from a store or restaurant and eat somewhere else [41]. Even before the pandemic, take-out food consumption has been on the rise worldwide in recent decades. In the UK, 21% of adults and children ate take-out meals once a week or more, and these consumption patterns were investigated similarly in other countries such as Europe, the USA, and Australia [42].

Janssen et al. [42] examined the strongest determinants of out-of-home food (takeout and fast food which are cooked out of home) with narrative analysis, and they found that density of food outlets and deprivation within the built environment were effective factors. During the pandemic, the inability to eat in stores with strengthening social distancing may have caused deprivation and increased numbers of take-out orders. Chenarides et al. [43] investigated impacts of COVID-19 on Qatar's food consumption, and they found that 74.9% of respondents were scared of COVID-19 and 66.3% felt unsafe participating in grocery pick-up or even delivery services. The COVID-19 pandemic and dining restrictions to prevent human contact deleteriously affected the food service industry. Byrd et al. [15] investigated customers' risk perceptions about food, restaurants, and food packaging during the pandemic and found that customers were concerned about contacting the virus from various types of food, including restaurant food. According to Byrd et al. [15],

customers were most worried about infecting COVID-19 from food served in restaurants, but least concerned about virus infection from food. Customers may have perceived that food is more harmful to eat directly because of the exposure of food to more employees and customers and the surfaces they touch, compared to delivered or packaged restaurant food. Further, customers were less concerned about COVID-19 infection in food delivered at restaurants rather than third-party delivery services [15]. Therefore, they prefer to consume food from take-out services to reduce viral infection through multiple stages of contact by a third party during the pandemic.

### 2.3. The Theory of Planned Behavior (TPB)

The Theory of Planned Behavior (TPB) [44] model is used to understand consumer behavior as an extension of the Theory of Reacted Action (TRA) [45]. Rational behavior theory has been used in many studies related to consumer behavior since its presentation, but it has shown a limitation that every behavior of consumers is hard to explain using only two factors, which are the attitude suggested in rational behavior theory and subjective norms. Therefore, the Theory of Planned Behavior model was established as an alternative model for these limitations [46]. The TPB model assumes that an individual's intention of action is determined by attitude, subjective norm, and perceived behavioral control [44]. The first factor, attitude (behavioral belief), is a degree or evaluation of favorable or unfavorable emotions for an action [45], and once formed, attitudes for a particular action tend to persist without changing [47]. The second factor, subjective norm (normative belief), refers to the tendency of people to expect specific behavior from an individual and the personal motivation to do so [44] In other words, when people who are important to themselves say that they have to do a specific action, the individual's intention for that action is formed even greater [47]. The last factor, perceived behavioral control (control belief), is the ability to overcome an individual's internal and external restrictions on a particular behavior and to perform an intended behavior; resources or opportunities must be supported [44].

The TPB model has increased explanatory power for behavioral intentions through more advanced behavioral control factors, and can be said to be one of the best established models to explain individual decision making and behavior [48,49]. It has been used to explain customers' behavior and behavioral intentions in several studies related to the hospitality industry, and extended planning behavioral theory models have also been used in several studies to improve customers' ability to predict behavioral intentions. Swine flu (2009 H1N1), an infectious disease which occurred in 2009, negatively affected the hospitality industry by reducing outbound tourism. Lee et al. (2012) [50] used the expanded Theory of Planned Behavior model to study the effect of swine flu on overseas travel. Suh et al. [51] conducted a study using an extended model that added factors such as trust and past experience to the original Theory of Planned Behavior model, and the results showed that all of the customers' attitudes, subjective norms, and perceived behavior control factors had direct or indirect effects. Kim et al. [39] used modified TPB to understand customers' behavioral intentions for drone food delivery services in relation to COVID-19, and found that customers' perceived innovation had a positive effect on attitudes, subjective norms, and perceived behavioral controls. Foroudi et al. [12] applied a belief model (actional beliefs, normative beliefs, controlling beliefs) to the relationship between customers' perception of shock of COVID-19 and their desire to visit restaurants in the future, even during the pandemic period. In addition, among these expectations, negative expectations do not affect restaurant visits in the future, and positive expectations increase the desire for restaurant visits.

### 2.4. Research Question

Based on the literature review, the following research questions were formed to solve the aims of this study:

RQ 1: There will be a gap in the degree of customers' dining-out activities before and after COVID-19 by type of dining out (visiting restaurants, using delivery services, or using take-out services).

RQ 2: Depending on the type of dining out (visiting restaurants, using delivery services, or using take-out services), there will be differences in the influence relationship between customers' attitude, subjective norm, perceived behavioral control, and behavioral intention during the pandemic.

In this study, as shown in Figure 1, the proposed conceptual model and how customers' dining-out activities have changed due to the influence of COVID-19 pandemic are presented. Therefore, first, the gap in the degree of dining-out activities of customers before and after the COVID-19 pandemic is investigated. Next, the study investigates the structural influence relationships of attitude, subjective norm, perceived behavioral control, and behavioral intention, which are the main variables of the TPB model, according to the three types of dining out activities: visiting restaurants, using delivery services, and using take-out services.

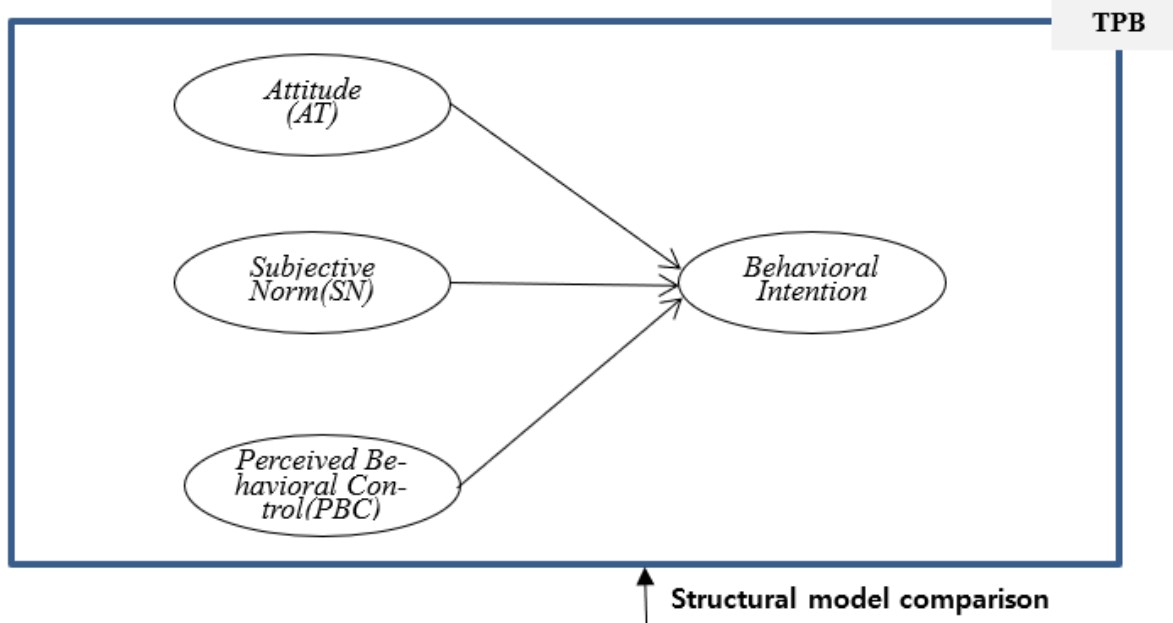

**Figure 1.** Proposed conceptual model.

### 3. Methods

*3.1. Sample and Data Collection*

In this study, a pilot survey was conducted on a total of 15 adults (20 s to 60 s) consisting of 5 graduate students majoring in food service management and 10 other general adults through e-mail. After the pilot survey, the final questionnaire was completed by adjusting the contents and the number of questions, and the completed questionnaire was distributed for about two months from 1 September to 31 October 2020 to general customers aged 20 or older who have experience in dining out. In consideration of the COVID-19 situation, an online survey method was used to distribute and collect survey link addresses through respondents' SNS and e-mail. Of the 500 distributed questionnaires, 468 (93.6%) were

recovered, the unfaithful responses were removed, and a data analysis was performed on the final 429 (85.8%) effective samples.

### 3.2. Research Instrument

In this study, customers' dining-out activities during the COVID-19 pandemic were classified into three categories: visiting restaurants, using delivery services, and using take-out services. After the COVID-19 pandemic, the frequency and expenditure of visiting restaurants per household decreased, while the frequency and expenditure of food delivery and take-out increased [52]. In this study, the Theory of Planned Behavior (TPB) model was applied to understand each customer's attitude, subjective norms, perceived behavioral control, and behavioral intentions about the three dining-out activities. Attitude is the first determinant of behavioral intention in the TPB model, meaning 'positive or negative evaluation of a specific behavior' [44], and in this study, a total of three questions, 'whether you think positively', 'whether you think it's valuable', and 'whether you think it's necessary', were measured for each of the three dining-out activities previously defined. Subjective norm is defined as the second determinant of behavioral intention in the TPB model as 'social pressure perceived not to perform or perform behaviors' [44], and refers to the opinions of close or important people influencing an individual's decision making. In this study, customers' subjective norms for each of the three dining-out activities were measured in three items: 'Surrounding (family, acquaintance, media, etc.) supports, recommends, and agrees'. Perceived behavioral control can be said to be 'perceived ease or difficulty in performing behavior', and evaluates how well behavior related to a specific situation can be performed and controlled [44,53]. In this study, how well individual behavior is controlled and prevented during the COVID-19 pandemic was measured with four criteria: hand washing, wearing a mask, refraining from unnecessary going out and meetings, and refraining from contacting others. The behavioral intention for dining-out activities was defined as the intention to do each of the three dining-out activities within the next six months in the pandemic situation, and three criteria of intention to use, plan to use, and frequency of use were measured (Table A1). South Korea faced the first pandemic of COVID-19 in February 2020, and the first death from COVID-19 infection occurred in the same month [54]. Therefore, this study measured the frequency of each of the three dining-out activities as of 1 February 2020, when COVID-19 began in earnest, to understand the difference in the three eating-out behaviors of customers before and after the COVID-19 pandemic. Multiple items with a five-point scale ranging from (1) 'strongly disagree (or rarely visit)' to (5) 'strongly agree (or very often visit)' were used to evaluate every construct.

### 3.3. Analysis

Statistical analysis of the collected data was conducted using SPSS 21.0 for Windows and AMOS 21.0. Frequency analysis was performed to understand the demographic characteristics of customers, and a paired t-test and explore analysis were conducted to compare the differences in the three dining-out activities of customers before and after COVID-19(RQ1). Confirmatory factor analysis, reliability analysis, and correlation analysis were performed to review the reliability and validity of the three dining-out activities (visiting restaurants, using delivery services, and using take-out services) and major variables (attitude, subjective norms, perceived behavioral control, and behavioral intentions), and both research questions were verified by structural equation modeling (SEM).

## 4. Results

### 4.1. Demographics of Respondents

The demographic profile of the respondents is shown in Table 1. Of the 429 respondents, 46.9% were female and 53.1% were male. A total of 82.7% of the respondents are in the age group 25–54 years. With regard to education, 73.0% obtained a college degree, 21.2% had degrees from graduate or higher, and 5.8% graduated from high school or less. Monthly

household income showed that 25.9% earned between 5,010,000–6,500,000 won (KRW), followed by between 3,510,000–5,000,000 won (KRW) (23.8%), 2,010,000–3,500,000 won (KRW) (19.6%), 8,010,000 won or more (14.0%), 6,510,000–8,000,000 won (KRW) (10.5%), and 2,000,000 won (KRW) or less (6.3%).

**Table 1.** Demographics of the participants (N = 429).

| Variables | Item | N | % | Variables | Item | N | % |
|---|---|---|---|---|---|---|---|
| Gender | Female | 201 | 46.9 | Education level | High school or less | 25 | 5.8 |
| | | | | | Bachelor's degree | 313 | 73.0 |
| | Male | 228 | 53.1 | | Graduate degree or over | 91 | 21.2 |
| Age | 20–24 | 18 | 4.2 | Monthly household income (KRW) | Less than 2,000,000 | 27 | 6.3 |
| | 25–34 | 116 | 27.0 | | 2,010,000–3,500,000 | 84 | 19.6 |
| | 35–44 | 120 | 28.0 | | 3,510,000–5,000,000 | 102 | 23.8 |
| | 45–54 | 119 | 27.7 | | 5,010,000–6,500,000 | 111 | 25.9 |
| | 55–64 | 50 | 11.7 | | 6,510,000–8,000,000 | 45 | 10.5 |
| | 65–74 | - | - | | 8,010,000 or more | 60 | 14.0 |
| | 75 or older | 6 | 1.4 | | | | |
| Family member | Alone | 50 | 11.7 | | | | |
| | Spouse | 38 | 8.9 | | | | |
| | Spouse/children | 207 | 48.3 | | | | |
| | Parent | 96 | 22.4 | | | | |
| | Parent/Spouse/children | 10 | 2.3 | | | | |
| | Children | 8 | 1.9 | | | | |
| | Others | 20 | 4.7 | | | | |

*4.2. Mean Differences between Pre- and Post-COVID-19 Outbreak*

In a first step, explore analysis was conducted to determine how the three dining-out activities of customers before and after the COVID-19 pandemic changed (Figure 2). As the result, regarding visiting restaurants, it was found that the median was 4.0 and the interquartile range was 1 (3–4) before the pandemic, while after the pandemic, the median was 2.0 and the interquartile range was 2 (1–3), indicating that after the pandemic, the visiting of restaurants has decreased significantly. In the case of using delivery services, the interquartile range before and after the pandemic was 2 (2–4), and the median showed a slight difference between before the pandemic (3.0) and after the pandemic (4.0). Regarding the use of take-out services, the interquartile range appeared as 1 (2–3) before the pandemic and 2 (2–4) after the pandemic, indicating a slight increase in the degree of use, and the median showed a slight increase in the degree of use before (2.0) and after (3.0) the pandemic.

In addition, before and after the COVID-19 pandemic, a paired t-test was performed to analyze the average difference between each of the three dining-out activities (Table 2). There were significant differences before and after the pandemic in all three types of dining-out activities, and regarding visiting restaurants, the difference between before (mean: 3.32) and after (mean: 1.97) was 1.34, indicating that before the pandemic was higher. In the case of using delivery services, there was a difference of −0.34 before the pandemic (mean: 2.94) and after the pandemic (mean: 3.28), and using take-out services, there was a difference of −0.32 before the pandemic (mean: 2.46) and after the pandemic (mean: 2.78), meaning that in both cases, it was found to have increased after the COVID-19 pandemic.

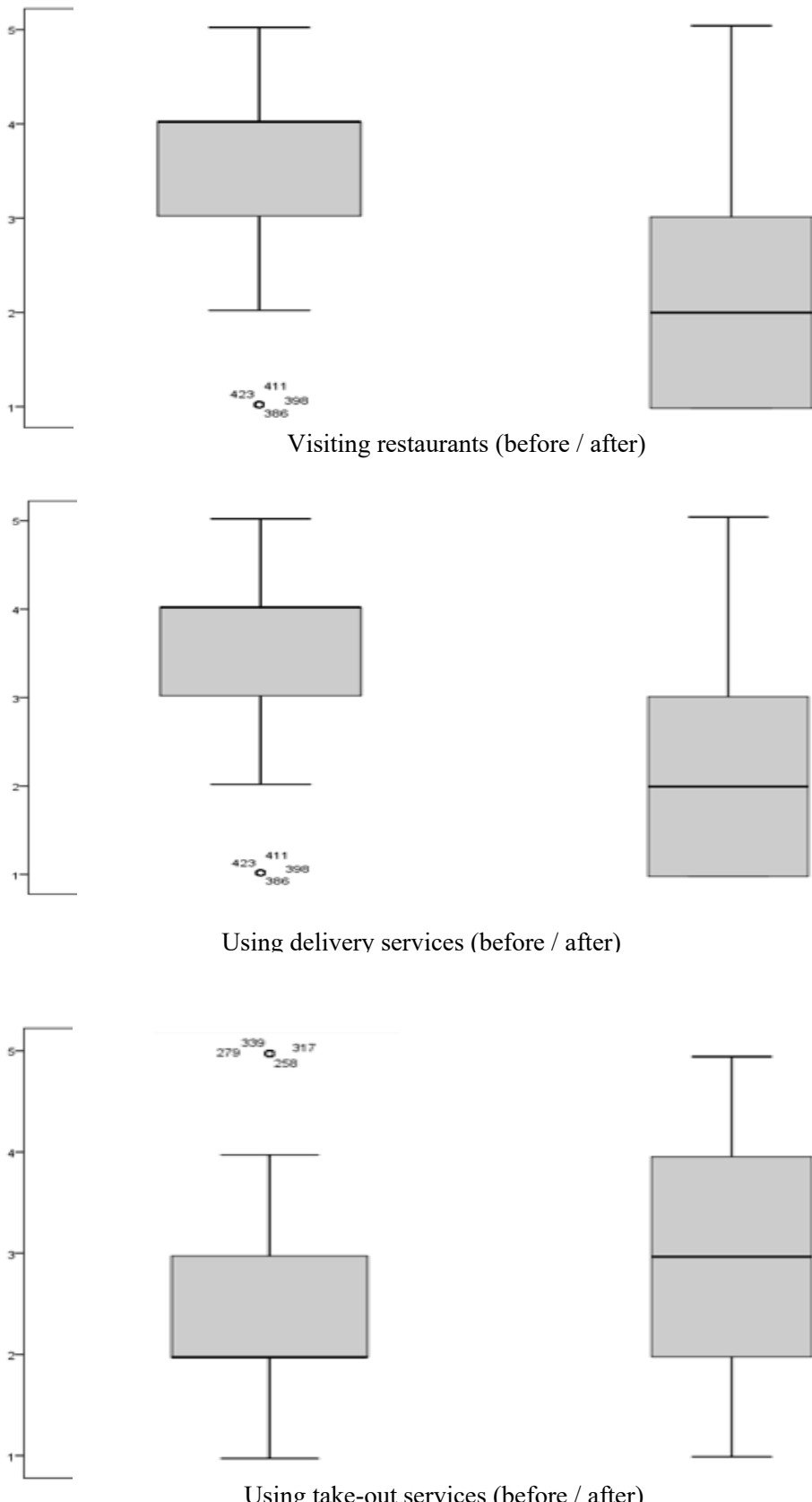

**Figure 2.** Box-plot distribution of dining-out experience before and after the COVID-19 pandemic.

**Table 2.** Mean differences between Pre- and Post-COVID-19 outbreak.

| Dining Out Types | Degree of Use | | | *t*-Value |
| --- | --- | --- | --- | --- |
| | Before | After | Difference | |
| visiting restaurants | 3.32 (1.04) | 1.97(0.94) | 1.34 | 20.865 ** |
| using delivery services | 2.94 (0.99) | 3.28(1.16) | −0.34 | −6.738 ** |
| using take-out services | 2.46 (1.06) | 2.78(1.13) | −0.32 | −5.676 ** |

** *p*< 0.01.

### 4.3. Measurement Model

Table 3 shows the results of the confirmatory factor analysis (CFA). The results of the CFA for the three types, which include visiting restaurants, using delivery services, and using take-out services, indicated that the overall fit of the measurement model was statistically satisfactory (visiting restaurants: $\chi^2$ = 199.200, df = 59, $\chi^2$/df = 3.376, GFI = 0.935, NFI = 0.952, IFI = 0.966, CFI = 0.966, TLI = 0.955, and RMSEA = 0.075; using delivery services: $\chi^2$ = 184.829, df = 59, $\chi^2$/df = 3.133, GFI = 0.938, NFI = 0.947, IFI = 0.963, CFI = 0.963, TLI = 0.951, and RMSEA = 0.071; and using take-out services: $\chi^2$ = 148.305, df = 59, $\chi^2$/df = 2.514, GFI = 0.950, NFI = 0.957, IFI = 0.974, CFI = 0.974, TLI = 0.965, and RMSEA = 0.059). All of the factor loadings were equal to or greater than 0.520 for visiting restaurants, 0.528 for using delivery services, and 0.532 for using take-out services. All standardized loadings were significant ($p$ < 0.01). This result supported convergent validity of the construct measures. In addition, all Cronbach's alpha values were from 0.813 to 0.941, exceeding the 0.6 recommended value, ensuring the construct reliability [55]. Table 3 shows the specific variables used in this study, along with their standardized factor loadings.

**Table 3.** Confirmatory factor analysis for measurement items.

| Construct and Scale Item | Standardized Loading | | |
| --- | --- | --- | --- |
| | Visiting Restaurants | Using Delivery Services | Using Take-Out Services |
| Attitude | | | |
| ATT1 | 0.862 | 0.864 | 0.849 |
| ATT2 | 0.896 | 0.761 | 0.792 |
| ATT3 | 0.853 | 0.777 | 0.829 |
| Subjective Norm | | | |
| SN1 | 0.872 | 0.841 | 0.839 |
| SN1 | 0.912 | 0.905 | 0.912 |
| SN1 | 0.891 | 0.796 | 0.789 |
| Perceived Behavioral Control | | | |
| PBC1 | 0.520 | 0.528 | 0.532 |
| PBC2 | 0.680 | 0.701 | 0.698 |
| PBC3 | 0.799 | 0.799 | 0.795 |
| PBC4 | 0.831 | 0.813 | 0.818 |
| Behavioral Intention | | | |
| BI1 | 0.914 | 0.933 | 0.898 |
| BI2 | 0.953 | 0.925 | 0.949 |
| BI3 | 0.868 | 0.813 | 0.861 |
| Model fit | $\chi^2$ = 199.200, df = 59, $\chi^2$/df = 3.376, GFI = 0.935, NFI = 0.952, IFI = 0.966, TLI = 0.955, CFI = 0.966, RMSEA = 0.075 | $\chi^2$ = 184.829, df = 59, $\chi^2$/df = 3.133, GFI = 0.938, NFI = 0.947, IFI = 0.963, TLI = 0.951, CFI = 0.963, RMSEA = 0.071 | $\chi^2$ = 148.305, df = 59, $\chi^2$/df = 2.514, GFI = 0.950, NFI = 0.957, IFI = 0.974, TLI = 0.965, CFI = 0.974, RMSEA = 0.059 |

### 4.4. Structural Equation Modeling Results

Covariance-based Structural Equation Modeling (CB-SEM) was employed to examine the relationships among attitude, subjective norm and behavioral intention according to the three types of dining-out activities. If the research objective is theory testing and confirmation, then the appropriate method is CB-SEM [56,57].

As is indicated in Table 4, Figure 3, first, when visiting a restaurant during the pandemic, customers' attitude ($\beta$ = 0.399, $p$ < 0.01) and subjective norm ($\beta$ = 0.269, $p$ < 0.01) had a positive effect on behavioral intention, while perceived behavioral control had a negative effect on restaurant visit intention ($\beta$ = −0.129, $p$ < 0.01). Second, when using delivery services, customers' attitude ($\beta$ = 0.582, $p$ < 0.01) and subjective norm ($\beta$ = 0.171, $p$ < 0.01) had a positive effect on behavioral intention, but perceived behavioral control had no significant effect on behavioral intention ($\beta$ = −0.037, $p$ > 0.05). Third, when using take-out services, customers' attitude ($\beta$ = 0.430, $p$ < 0.01) and subjective norm ($\beta$ = 0.275, $p$ < 0.01) had a positive effect on behavioral intention, while perceived behavioral control had a negative effect on behavioral intention ($\beta$ = −0.099, $p$ < 0.05).

**Table 4.** Results of hypothesis tests: TPB during the pandemic.

| Hypotheses | Path Coefficient | | | | | |
|---|---|---|---|---|---|---|
| | Visiting Restaurants (a) | | Using Delivery Services (b) | | Using Take-Out Services (c) | |
| | $\beta$ | *t*-Value | $\beta$ | *t*-Value | $\beta$ | *t*-Value |
| ATT → BI | 0.399 | 6.304 ** | 0.582 | 9.462 ** | 0.430 | 6.395 ** |
| SN → BI | 0.269 | 4.170 ** | 0.171 | 3.028 ** | 0.275 | 4.264 ** |
| PBC → BI | −0.129 | −2.737 ** | −0.037 | −0.862 | −0.099 | −2.170 * |
| SMC [a] | 0.455 | | 0.486 | | 0.423 | |
| Model fit | $\chi^2$ = 199.200, df = 59, $\chi^2$/df = 3.376, GFI = 0.935, NFI = 0.952, IFI = 0.966, TLI = 0.955, CFI = 0.966, RMSEA = 0.075 | | $\chi^2$ = 184.829, df = 59, $\chi^2$/df = 3.133, GFI = 0.938, NFI = 0.947, IFI = 0.963, TLI = 0.951, CFI = 0.963, RMSEA = 0.071 | | $\chi^2$ = 148.305, df = 59, $\chi^2$/df = 2.514, GFI = 0.950, NFI = 0.957, IFI = 0.974, TLI = 0.965, CFI = 0.974, RMSEA = 0.059 | |

Note: * $p$ < 0.05, ** $p$ < 0.01 a: Squared Multiple Correlations. ATT: Attitude, BI: Behavioral intention, SN: Subjective norm, PBC: Perceived behavioral control.

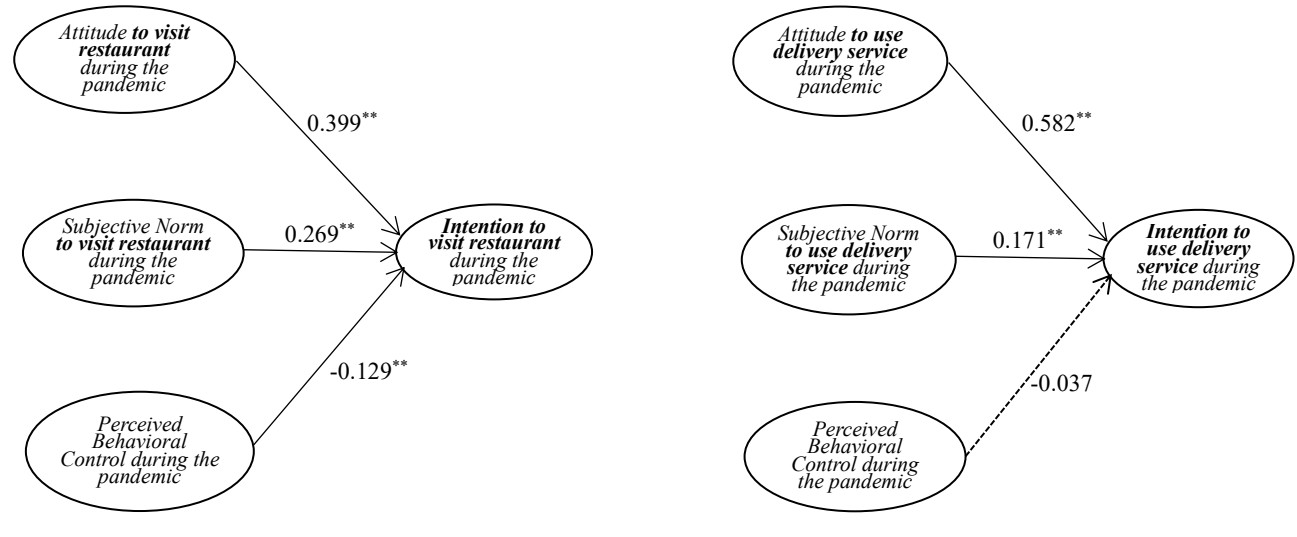

-Visiting restaurants(a)-                    -Using delivery services(b)-

**Figure 3.** *Cont.*

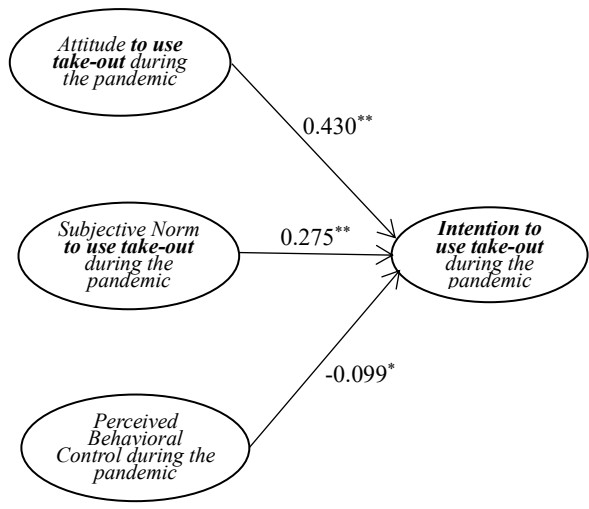

-Using take-out services(c)-

*p<0.05 , **p<0.01

Solid lines indicate significant paths(β), while dotted line indicates non-significant path

**Figure 3.** Results of the structural model and path coefficients.

## 5. Discussion and Conclusions

The lifestyle of customers has changed a lot due to the COVID-19 pandemic, and among them, dining-out activity has been directly affected. Therefore, this study investigated how customers' dining-out activities changed before and after the COVID-19 pandemic. First, among the three types of dining-out activities (visiting restaurants, using delivery services, and using take-out services), the biggest change before and after the pandemic was the increase in the use of delivery services. Although the number of customers visiting restaurants has decreased rapidly due to the pandemic, the use of delivery services with relatively low perceived risk has increased. Second, as a result of applying the Theory of Planned Behavior model to understand changes in customers' behavior due to COVID-19 pandemic, there were differences in the influence relationship of attitude, subjective norm, and perceived behavioral control, and behavioral intention for each of the three dining-out activities. In the case of visiting restaurants, if customers have a positive attitude toward visiting themselves, or a positive subjective norm is formed by the responses of people around them, intention to eat in the restaurant increases, while perceived behavioral control related to hygiene or quarantine compliance negatively affected the intention to eat in the restaurant. On the other hand, in the case of using delivery services, only attitude and subjective norm were found to have a positive effect on behavioral intention. In the case of using take-out services, results similar to those of visiting restaurants were found. This can be said to be because, by using take-out services, customers have to visit restaurants in person and use the service, making customers aware of the perceived risk being relatively high.

## 6. Implications and Future Research Suggestions

### 6.1. Academic Implications

First, in the pandemic situation caused by COVID-19, previous studies on customers' dining-out activities focused only on one behavior, such as visiting restaurants [30], using delivery services [37], or using take-out services [15]. However, unlike previous studies, this study is of significant contribution in that it compared and analyzed customers' attitudes, subjective norms, perceived behavioral control, and behavioral intention to various dining-out activities, such visiting restaurants, using delivery services, and using take-out

services, using the Theory of Planned Behavior model. The second contribution is that this study structurally analyzed the influence relationship between customers' attitudes, subjective norms, perceived behavioral control, and behavioral intention through structural equation modeling (SEM). As the result, it was found that, among the three dining-out activities, visiting restaurants and using take-out services showed the same structure of influence relationship. Regarding visiting restaurants and using take-out services, perceived behavioral control has negatively affected behavioral intention, while in the case of using delivery services, perceived behavioral control has not significantly affected behavioral intention. This result means that customers must move out and eat out on their own to visit restaurants and to use take-out services, which strengthens perceived behavioral control such as washing hands and wearing masks, thereby lowering their behavioral intention to dine out. According to Ajzen [44], people intend to perform an action according to the degree to which they believe they have control over an action, so under the pandemic, customers' control behaviors related to dining-out activities, such as compliance with personal hygiene, unnecessary refraining from going out, and refraining contact with others, and it can be said that customers' perceived behavioral control influenced the intention to behave [58,59]. Third, in this study, the types of dining-out activities were subdivided and the differences in each subdivided dining-out activity before and after the pandemic were compared and analyzed, which is also a significant contribution. It is difficult to find a previous study comparing the gap in dining-out activities before and after the pandemic, and only a few studies measured the change in the degree of customer use for only one dining-out activity after the COVID-19 pandemic [28,38,43].

### 6.2. Managerial Implications

First, this study confirmed that after the outbreak of COVID-19, the degree of visiting restaurants was lower than before COVID-19, and through this result, it was found that risk factors related to infectious diseases had the greatest influence on visit restaurants among the three types of dining-out activities (see Figure 2, Table 2). According to a study related to COVID-19, many customers think it is safer to use take-out services or cook at home than to eat at restaurants [26], and several studies have stated that restaurant sales have fallen by more than 85% since the COVID-19 pandemic [25], and that the intention to visit restaurants and the actual degree of visits to restaurants have decreased significantly [30–32]. Therefore, governments should establish proper hygiene rules and systematic measures to adjust the level of distancing according to changes in the COVID-19 environment, and the food service industry should thoroughly comply with the quarantine rules so that customers can participate in stable dining-out activities as soon as possible. Second, delivery services, which began to grow in the food service industry before the outbreak of COVID-19, are expected to occupy a more important position in dining-out activities in the future due to COVID-19. Customers who know that non-contact with others lowers the risk of infection have come to prefer food delivery services [36]. As a result, many restaurant companies have further activated delivery services, and actual delivery food consumption has increased by 79 percent in South Korea [21]. As delivery services occupy an important position under the pandemic, the delivery industry should follow quarantine rules more thoroughly and come up with measures to maintain the taste and freshness of food so that customers can participate in dining-out activities with confidence.

### 6.3. Limitations and Future Research Suggestions

This study has several limitations despite a great deal of academic and managerial implications, as mentioned above. First, it was intended to analyze the differences in dining-out activities of various age groups during the pandemic, but the survey method was conducted online due to social distancing, so the number of samples for groups over the age of 55 was small, so it was not possible to analyze the difference in dining-out activities between age groups. Second, in this study, the factors influencing customers' intentions to participate in dining-out activities were limited to only three factors: attitude,

subjective norms, and perceived behavioral control, and the influence relationship on more diverse factors could not be revealed. Therefore, by overcoming the limitations of this study, research that considers more diverse demographic and social factors and various influencing factors should be conducted in the future.

**Author Contributions:** Conceptualization, B.S. and H.M.; methodology, H.M.; formal analysis, H.M.; writing—original draft preparation B.S., S.K. and H.M.; writing—review and editing, B.S., S.K. and H.M.; supervision, B.S.; project administration, B.S., S.K. and H.M. All authors have read and agreed to the published version of the manuscript.

**Funding:** This research received no external funding.

**Institutional Review Board Statement:** Not applicable.

**Informed Consent Statement:** Not applicable.

**Data Availability Statement:** The dataset used in this research are available upon request from the first author. The data are not publicly available due to restrictions i.e., privacy or ethical.

**Conflicts of Interest:** The authors declare no conflict of interest.

## Appendix A

**Table A1.** Definition of Variables.

| Variable | Measure | References |
|---|---|---|
| Attitude | whether you think it's valuable | [53] |
| | whether you think positively | [50] |
| | whether you think it's necessary | [51] |
| Subjective Norm | Surrounding me — that I eating out activity. support | [53] [50] |
| | recommend | [51] |
| | agree | |
| Perceived Behavioral Control | hand washing | [20] |
| | wearing a mask | [58] |
| | refraining from unnecessary going out and meetings | [51] |
| | refraining from contacting others | |
| Behavioral Intention | within the next six months intention to use | [53] |
| | plan to use | [50] |
| | frequency of use | [51] |

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
