# Peer review of "Estimating the Impact of COVID-19 Pandemic on Customers’ Dining-Out Activities in South Korea"

_sustainability, doi:10.3390/su14159408_

Round 1
Reviewer 1 Report
The covid-19 pandemic changed the architectures of the world we know. The way of functioning in the workplace, traveling, contact with friends and even eating meals, changed significantly during the covid-19 pandemic. In this study, the authors compared the degree of customer use to three dining activities before and after COVID-19, and used the planned behavior model (CBA) theory to compare and analyze the structural relationship between customer attitudes, subjective norm, perceived behavioral control, and behavioral intentions for each of the three activities related to eating out.
The article presents in an interesting way how the habits of consumers have changed. below I present my comments for consideration by the authors.
2. Literature Review and Research Question
Line 80 - please insert the actual data.
Are there any statistics known to the author about how many people contracted coranovirus while eating meals prepared by people infected with coranovirus?
Before the covid-19 pandemic in South Korea, what percentage of people ate away from home?
How widespread is the use of drones in providing food to customers?
What demographic and social factors make people more likely to choose to eat out?
Could the figures on page 9 be combined into one?
How can customers find out that food delivery is safe?
Reviewer 2 Report
As in the attached files

Reviewer 3 Report
Abstract
I suggest the authors rewrite the abstract. Using the following structure:
- concisely state the objective of the study.
- theory used.
-methodology (data collection, population) and analysis
-originality of the paper
-Practical implications
Introduction
Please revise the introduction according to the following structure:
1. Introduce the topic i.e., – define the topic, explain them and use examples to explain their importance. Add some statistics on the market value.
2. General text about the topic (one para), focus on its relevance and importance for the industry, practice and theory
3. Now briefly inform readers the studies conducted. xx and xx studies have been organised so far. One para on what has been done so far on the topic
4. Now talk about research gaps- What are the key research gaps and why they are important or need to be addressed now. Clearly present 3-4 research gaps on this topic
5. Present the focus of the current study. What are its RQs and details on method – briefly say about data, country, context and theory
6. Novelty and contribution of the study
7. Structure of the paper
Additionally, more references can be added. Currently, there are arguments that are not supported by references.
Consider adding the following reference for this sentence
1) “ To prevent the rapid and widespread spread of COVID-19, countries 30 around the world implemented travel restrictions, street or city closures, and applied 31 masks and social distancing to prevent infection” - Sharma, S., Singh, G., Sharma, R., Jones, P., Kraus, S., & Dwivedi, Y. K. (2020). Digital health innovation: exploring adoption of COVID-19 digital contact tracing apps. IEEE Transactions on Engineering Management.
2) “Pandemics are expected to continue over the long term, which will have a profound 57 impact on customers’ consumption behaviors..” - Singh, G., Aiyub, A. S., Greig, T., Naidu, S., Sewak, A., & Sharma, S. (2021). Exploring panic buying behavior during the COVID-19 pandemic: a developing country perspective. International Journal of Emerging Markets.
3) “Researchers have conducted re-60 search on COVID-19 in the field of hospitality and dining out industry, but customer re-61 search is still insufficient” – this sentence can also be supported by this reference - Sharma, S., Singh, G., Ferraris, A., & Sharma, R. (2022). Exploring consumers’ domestic gastronomy behaviour: a cross-national study of Italy and Fiji. International Journal of Contemporary Hospitality Management.
4) “It has been used to explain customers’ behavior and behavioral intentions in several 243 studies related to the Hospitality industry, and extended planning behavioral theory 244 models have also been used in several studies to improve customers’ ability to predict 245 behavioral intentions” – consider adding the following studies that have used TPB –
Singh, G., Sharma, S., Sharma, R., & Dwivedi, Y. K. (2021). Investigating environmental sustainability in small family-owned businesses: Integration of religiosity, ethical judgment, and theory of planned behavior. Technological Forecasting and Social Change, 173, 121094.
Sharma, R., Singh, G., & Sharma, S. (2021). Competitors' envy, gamers' pride: An exploration of gamers' divergent behavior. Psychology & Marketing, 38(6), 965-980.
Sharma, S., Singh, G., & Sharma, R. (2021). For it is in giving that we receive: Investigating gamers’ gifting behaviour in online games. International Journal of Information Management, 60, 102363.
Methodology
The methodology section is weak. Important details are missing. The authors need to provide more details about how the online data collection took place. Was convenience or random sampling used? Where was the online questionnaire hosted? SurveyMonkey? Was the pilot study conducted? Details about language translation?
The authors need to also describe how the data screening procedure? Unengaged respondents? Confirmation of normality? Multicollinearity issues?
Data analysis
Authors need to highlight whether CB-SEM or PLS-SEM was used. The justification for using either method needs to be highlighted. The following can be used as a guide to writing this section.
Limitations and future research direction
This section needs to be strengthened. The limitation of the cross-section design of this study needs to be highlighted. Generalizability issues? More interesting directions for future research can be proposed.
